# Colorectal Cancer and the Role of the Gut Microbiota—Do Medical Students Know More Than Other Young People?—Cross-Sectional Study

**DOI:** 10.3390/nu14194185

**Published:** 2022-10-08

**Authors:** Paulina Helisz, Grzegorz Dziubanek, Karolina Krupa-Kotara, Weronika Gwioździk, Mateusz Grajek, Joanna Głogowska-Ligus

**Affiliations:** 1Department of Epidemiology, Faculty of Health Sciences in Bytom, Medical University of Silesia in Katowice, 41-902 Bytom, Poland; 2Department of Environmental Health, Faculty of Health Sciences in Bytom, Medical University of Silesia in Katowice, 41-902 Bytom, Poland; 3Department of Public Health, Faculty of Health Sciences in Bytom, Medical University of Silesia in Katowice, 41-902 Bytom, Poland

**Keywords:** gastrointestinal cancer, gut microbiota, colorectal cancer, students, level of knowledge

## Abstract

(1) Background: Malignant neoplasms account for an increasing share of the disease burden of the world population and are an increasingly common cause of death. In the aspect of colorectal cancer, increasing attention is paid to the microbiota. According to current knowledge, the composition of gut microbiota in patients diagnosed with colorectal cancer significantly differs from the composition of microorganisms in the intestines of healthy individuals. (2) Material and methods: The survey included 571 students from the three universities located in Silesia. The research tool was an original, anonymous questionnaire created for the study. The ratio of correct answers to the total number of points possible to obtain was evaluated according to the adopted criteria (≤25%—very low level of knowledge; >75%—high level of knowledge). (3) Results: From the questions about the gut microbiota, the subjects scored an average of six points (SD ± 1.31) out of nine possible points. Statistical analysis showed differences between the number of correct answers among students of the Medical University of Silesia and the University of Silesia (*p* = 0.04, *p* < 0.05). On the other hand, in the field of colorectal cancer, the respondents scored on average four points (SD ± 2.07) out of eight possible. Statistical analysis showed significant differences between the ratio of correct answers and the respondent’s university affiliation (*p* < 0.05). Both age and place of residence did not positively correlate with knowledge level (*p* = 0.08 NS). In contrast, chronic diseases were found to have a significant effect on the amount of information held by the students surveyed (*p* < 0.05). (4) Conclusions: The level of knowledge of the surveyed students of the Silesia Province is unsatisfactory. The higher awareness among the students of medical universities results from the presence of issues related to microbiota and CRC in the medical educational content. Therefore, there is a need to consider the introduction of educational activities in the field of cancer prevention, including CRC, especially among non-medical university students.

## 1. Introduction

Malignant neoplasms account for an increasing share of the disease burden of the world’s population. In the structure of causes of death in the world as of 2019, malignant neoplasms ranked second, just after cardiovascular diseases [1]. Cancer is one of the leading causes of death for people around the world. According to the latest statistics from the International Agency for Research on Cancer (IARC), there will be more than 19 million cases of cancer in 2020 [2]. In both men and women, colorectal cancer (CRC) is the third most common malignancy [2,3,4]. According to recent statistics, the incidence of colorectal cancer among Europeans is higher in men (*n* = 191,053) than in women (*n* = 150,366) [5]. Accordingly, among the most important public health tasks is cancer prevention. Prevention methods focus primarily on the formation of a healthy lifestyle, which should be characterized by, among other things, a well-balanced diet, and physical activity, as well as limiting exposure to environmental risk factors that exhibit carcinogenic effects [4,6,7]. Microbiota-related issues are also receiving increasing attention in the prevention aspect of CRC. According to the current state of knowledge, the composition of the intestinal microbiota in patients diagnosed with colorectal cancer differs significantly from that shown in healthy individuals [8,9]. In addition, dysbiosis of the human terminal gastrointestinal tract may be one of the risk factors for the development of CRC. It has been shown that the gut microbiota may play an important role in both the progression and the body’s response to the treatment of this type of cancer [5,6,7,8]. Research is also underway to develop a highly sensitive biomarker of gut microbiota composition that can be used in screening for CRC [8,10].

### Gut Microbiota and CRC

The gut microbiota refers to the microorganisms residing in the large intestine, which include bacteria, eukaryotes, viruses, and archaeons. It consists of 10^13^ to 10^14^ microbes containing a total of 100 times more genes compared to the human genome [11,12]. A properly colonized gut microbiota plays an important role in the human body. Some of its most important aspects include: acting as a protector against pathogens, shaping the intestinal epithelium, or supporting host immunity. Microorganisms residing in the human large intestine also contribute to nutrient absorption and metabolism, as well as participate in the production of, for example, vitamins B and K [13,14].

Dysbiosis is referred to as changes in the composition of the gut microbiota, which contribute to the development of various diseases, e.g., irritable bowel syndrome, food allergies, asthma, and even neuropsychiatric disorders [15,16]. Causes of disorders in the human intestinal ecosystem include factors such as obesity, inadequate diet (high-fat, processed), use of antibiotics, exposure to heavy metals and pesticides, type of feeding during infancy (breast or milk-replacement mixture), as well as a mode of delivery via cesarean section [13,17]. Research emphasizes that intestinal dysbiosis can result in an increased risk of developing inflammation in the body [17,18].

Lifestyle is among the most important factors modulating the composition of the gut microbiota. According to current knowledge, dietary behavior is one of their main components. The way of eating, known as the “Western diet,” which is characterized by a high intake of processed foods, high in fat and rich in simple sugars, can lead to dysbiosis [19]. In contrast, a diet rich in vegetables, fruits, whole grains, and anti-inflammatory ingredients, such as olive oil, nuts, red wine, and coffee, provides health benefits by properly modulating the microorganisms populating the gut [19,20]. Available scientific evidence suggests that the Mediterranean diet can be considered a model of healthy eating, positively influencing the composition of the gut microbiota in humans, as well as their immune systems [20]. The assumptions of the world’s healthiest diet pay considerable attention to the regular consumption of vegetables, especially cruciferous vegetables such as broccoli, cabbage, and Brussels sprouts, due to the carotenoids, lycopene, folic acid, selenium, and B vitamins they contain. The importance of consuming fruits and whole-grain products, which are sources of dietary fiber, is also emphasized. Fish, nuts, and olive oil are also used in the Mediterranean diet. This is because omega-3 fatty acids, abundant in the aforementioned products, have valuable antioxidant properties that favorably slow down cell proliferation, angiogenesis, and inflammation, which, as a result, may favorably contribute to a reduced risk of cancer (including colon cancer) [21,22]. Research suggests that DHA (Docosahexaenoic acid) and EPA (Eicosapentaenoic acid) positively correlate with the microbial diversity of the host gut, by increasing the number of *Bifidobacteria* and *Lactobacillus* [22].

Proper colonic homeostasis plays a key role in terms of maintaining good health. When colonic homeostasis is disturbed, the metabolism of colonocytes is altered, which contributes to intestinal dysbiosis [23]. The appropriate composition of microorganisms in the human gastrointestinal tract, induces a normal immune response, preventing the settlement of pathogens. Scientific evidence indicates that the intestinal microbiota is considered an important “organ” that determines the proper function of the immune system via intestinal epithelial cells and intraepithelial lymphocytes [24]. Any disruption of the immune response contributes to the process of carcinogenesis, as well as decreases the effectiveness of anti-cancer therapy [24,25]. A healthy gut microbiome maintains an appropriate balance between the concentration of pro-inflammatory and anti-inflammatory cytokines, as well as between immune cells and IgA (Immunoglobulin A) secretion. The adequate diversity of microorganisms residing in the gastrointestinal tract, as well as a normal, intact mucosal barrier, is extremely important [26]. Dysbiosis in the colon is associated with an increased presence of anaerobic bacteria in the intestines, which consequently positively correlates with the incidence of chronic diseases such as irritable bowel syndrome, inflammatory bowel disease, and CRC [23].

The composition of the gut microbiota of a patient with CRC differs significantly from that of a healthy person. In people diagnosed with end-stage gastrointestinal cancer, bacteria such as *Fusobacterium nucleatum*, *Escherichia coli*, and *Bacteroides fragilis* [26]. The last bacterial strain contributes to the development of inflammatory bowel disease [12], which is classified as a factor that significantly increases the risk of CRC [22,27]. On the other hand, *F. nucleatum* induces the formation of precancerous lesions, which are intestinal adenomas, and contributes to their progression to cancer. In addition, scientific studies have shown the presence of the above bacterial strain in the intestinal tissues of patients diagnosed with early onset (<50 years of age). As indicated by literature data, people with diagnosed CRC are characterized by higher amounts of *Escherichia coli* in the intestine compared to healthy people. The pathomechanism of the development of CRC is complex, so it is not possible that one of the potentially pathogenic microorganisms contributes to the progression of cancer. In the case of the intestinal microbiota, the adverse effects of bacteria outweigh the beneficial effects of commensals, resulting in the induction of intestinal dysbiosis and its further consequences [13].

Taking this into account, in the present study we decided to conduct an awareness study of a population of students from the Silesian voivodeship (in Poland) regarding the potential impact of intestinal microbiota on the risk of colorectal cancer. The choice of the study population was not coincidental as it served to assess the knowledge of young people who, through their current behavior, lifestyle and diet, influence the composition of their intestinal microbiota, and thus determine the magnitude of their risk of future CRC. The inclusion of medical and non-medical students in the study was aimed at determining whether the knowledge gained by future doctors and health care professionals as part of the educational process has a real impact on a better awareness of the vital importance of microbiota in the incidence of colorectal cancer.

## 2. Materials and Methods

### 2.1. Study Organization and Eligibility Criteria

The survey included 605 students from universities located in Upper Silesia, and it was a descriptive cross-sectional study. The survey was conducted online from April to June 2022. The three most populous universities, in terms of the number of students, were included, according to data posted on the Upper Silesian Metropolitan Area website [28]. The research tool of the study was the author’s anonymous questionnaire, which was validated for reliability, correctness, and relevance. The responses to the same questions were checked for consistency. To assess the reproducibility of the results obtained with the used questionnaire, the value of the parameter κ (Kappa) was calculated for each question in the questionnaire—for 63.3% of the questions, a very good (κ ≥ 0.80) concordance of answers was obtained, while for 36.7% of the questions, a good (0.79 ≥ κ ≥ 0.60) concordance of methods was obtained. The final stage of the study was to conduct the actual test. The study assessing the level of knowledge does not require the approval of the Bioethics Committee, as it is not a medical experiment in light of the Act of 5 December 1996, on the professions of physician and dentist (Journal of Laws of 2011 No. 277. item 1634 as amended). In addition, the study was conducted by the provisions of the Declaration of Helsinki.

The exclusion criteria for the study were age <18 years and >26 years, as well as lack of student status. As a result, statistical analysis was carried out based on 571 questionnaires. The selection of respondents was non-random, as the snowball method was used. Participation in the survey was voluntary and respondents were assured of anonymity. For a sample of N_US_ = 283 (the University of Silesia in Katowice), N_PolSl_ = 181 (Silesian University of Technology), and N_SUM_ = 107 (Medical University of Silesia), the necessary sample size was calculated, depending on the higher education institution, using the formula for a finite population.

### 2.2. Study Procedure and Research Tool

The questionnaire consisted of a metric part, ten closed questions on knowledge, as well as two closed questions subjectively assessing the amount of information the subject possessed in the field of microbiota and colorectal cancer. The questionnaire addressed issues relating to the definition of intestinal microbiota, probiotic and prebiotic, as well as the meaning of the term “precancerous condition”. In addition, the questionnaire included questions relating to the symptomatology of colorectal cancer. To identify respondents, they were asked to answer questions regarding age, gender, place of residence, university affiliation, current body weight, and height.

### 2.3. Interpretation of the Tools Used

Based on the declared anthropometric values, a body mass index (BMI) was calculated and interpreted according to the World Health Organization [29]: underweight (<18.5 kg/m^2^), normal (18.5–24.9 kg/m^2^), pre-obesity (25.0–29.9 kg/m^2^), obese I° (30.0–34.9 kg/m^2^), obese II° (35.0–39.9 kg/m^2^), obese III° (>40 kg/m^2^). The student’s level of knowledge was assessed by points: 1- for a correct answer; 0- for an incorrect one. The final score was a component of the sum of correct solutions to the total number of questions on intestinal microbiota and colon cancer. The maximum number of points possible to obtain from the knowledge part was 25. The ratio of correct answers to the total number of points possible to obtain was evaluated according to the adopted criteria [30]:≤25%—very low level of knowledge,26–50%—low level of knowledge,51–75%—medium level of knowledge,>75%—high level of knowledge.

### 2.4. Statistical Compilation

The database and graphical results were compiled using MS Excel, while statistical analysis was performed using Statistica 13.3 (TIBCO Software Inc., Palo Alto, CA, USA). The Shapiro–Wilk test was used to assess the normality of the measurable variables. The variables analyzed were mostly qualitative in nature, so non-parametric tests were used to assess statistical significance. The nonparametric Kruskal–Wallis rank-sum ANOVA test was used to compare three or more independent groups, followed by multiple comparisons using the post-hoc test. The χ^2^ test was used to assess the knowledge of independent groups. Measurable data were presented by mean value and standard deviation. The strength of the stochastic relationship between non-measurable characteristics was carried out using the V-Cramer correlation coefficient. The level of statistical significance was considered to be *p* < 0.05. 

## 3. Results

The study included 571 respondents, the vast majority of whom were women. The mean age of the respondents was 22 years (SD ± 1.82). Students were asked to enter their body weight (in kilograms) and height (in centimeters) values into a form. Based on the declared data, a body mass index (BMI)-kg/m^2^ was calculated. Preliminary analysis in terms of numbers showed that more than half of the respondents have normal body weights. The answer to the question about current body weight as well as height was voluntary, so seven respondents (1%) chose to skip this part. Detailed characteristics of the study group are shown in Table 1.

### 3.1. Students’ Knowledge of Microbiota and CRC

#### 3.1.1. Microbiota

Of the questions on microbiota, students did best with the question relating to the most important factor determining the correct composition of microorganisms residing in the intestine, namely diet. The total number of correct answers to this question was 91% (*n* = 517), indicating a high level of knowledge. In the statistical analysis, there was no significant effect of the affiliation of the university where they studied versus the correct solution (*p* = 0.37; *p* > 0.05). The question of the definition of microbiota proved to be the most problematic. Less than one in four students demonstrated proficiency on the question (*n* = 220; 39%). Statistical analysis showed significant differences between the most proficient students of the Medical University of Silesia (SUM) compared to students of the University of Silesia (US) or the Silesian University of Technology (PolSl) (*p* = 0.001; *p* < 0.05). A detailed interpretation of the microbiota questions is shown in Table 2.

#### 3.1.2. Colorectal Cancer

Respondents were also asked whether normal gut microbiota composition correlates positively with an increased risk of colorectal cancer. According to 87% of SUM students surveyed, the composition of the intestinal microbiota may influence the development of colorectal cancer. In the case of students from the Silesian University of Technology, the percentage perceiving a causal relationship between the composition of the microbiota and the risk of colorectal cancer was 59%, while among students from the University of Silesia the percentage was only 53%.

Based on the data in Table 3, it can be deduced that more than half of the surveyed students from the Silesian region know the definition of precancerous colorectal conditions (*n* = 355; 62%). Statistical analysis showed a significant effect between university affiliation and the level of knowledge of colorectal cancer (*p* = 0.0001; *p* < 0.05). Medical school (SUM) students, for both the definition of precancerous colorectal conditions and the prevalence of intestinal cancer, showed the highest knowledge. In question one (presented in Table 3), the answer “I don’t know” was recorded in 155 (27%) cases (N_US_ = 85; N_PolSl_ = 60; N_SUM_ = 10), while in question two the number of respondents declaring lack of knowledge was higher, reaching 238 (42%) of respondents (N_US_ = 125; N_PolSl_ = 91; N_SUM_ = 22).

As part of this study, respondents were asked to answer a series of questions about the impact of gut microbiota on the human body. The students did best on questions about the composition of the gut microbiota versus the risk of contracting CRC (*n* = 537; 94%), as well as the impact of the microbiota on the immune system (*n* = 526; 92%). On average, respondents scored six points (SD ± 1.31) out of nine possible points. Statistical analysis showed significant differences between the number of correct answers given by students from the Medical University of Silesia and the University of Silesia (*p* = 0.04, *p* < 0.05). In contrast, there were no statistically significant differences between the answers given by students of the Silesian University of Technology and students of other universities in the Silesian province (*p* > 0.05 NS).

Respondents were also asked questions about the symptoms of CRC. For each symptom presented, respondents could select one of three possible options such as “yes”-it occurs in the course of CRC; “no”-it does not occur in the course of CRC and “don’t know.” The author’s questionnaire included a list of the following symptoms: diarrhea, constipation, vomiting, blood in the stool, lower gastrointestinal bleeding, anemia, and weight loss. Respondents scored an average of four points (SD ± 2.07) out of a possible eight. Less than 5% of students from the Medical University of Silesia (*n* = 5) and 1% of respondents from the University of Silesia (*n* = 4) achieved the maximum score. In the case of students from the Silesian University of Technology, none of the respondents obtained the maximum number of points. Respondents who declared they were educated at SUM scored an average of six points (SD ± 1.48), while respondents from the University of Silesia and Silesian University of Technology scored four points (SD ± 2.07; SD ± 2.12, respectively). Statistical analysis showed significant differences in the number of correct answers given and the university from which the respondents came (*p* = 0.0001; *p* < 0.05).

Figure 1 shows the responses in percentages. The summary does not include any missing responses (Figure 1). The most recognizable symptom of CRC was blood in the stool (85%) and bleeding from the lower gastrointestinal tract (79%). A lower percentage of respondents were able to identify such CRC symptoms as diarrhea (64%), weight loss (63%), and constipation (62%). The least frequent respondents indicated anemia (33%) and dizziness (27%).

### 3.2. Knowledge Level of Surveyed Students

Respondents’ answers were analyzed in terms of their level of knowledge regarding intestinal microbiota and colorectal cancer, taking into account the point scale described in the Materials and Methods section. Respondents, based on their answers, could score a maximum of 25 points (100%). The average score obtained was 16 (SD ± 4.01), indicating the average level of knowledge of students in the Silesian province regarding the issues under consideration.

Statistical analysis showed that one in four respondents had a high level of knowledge (*n* = 143; 25%) regarding intestinal microbiota and colorectal cancer, the highest percentage of which were students from the Silesian Medical University (*n* = 66; 46%). More than half of the respondents (*n* = 306; 54%) had an intermediate range of knowledge about the microbiota as well as CRC, while a low 19% of the respondents (*n* = 109). In contrast, the weakest score was obtained by 2% (*n* = 13) of respondents, the vast majority of whom were students at the Silesian University of Technology (*n* = 8) (Figure 2).

Statistical analysis showed a significant effect of university type on the level of the student’s knowledge (*p* = 0.0001; *p* < 0.05). The strength of the association of the analyzed statistical characteristics was medium (*Vc* = 0.3). Details of the respondent’s level of knowledge according to the university they attend are shown in Table 4.

The results of the self-assessment of the knowledge of surveyed students from the Silesian province in the field of microbiota and colorectal cancer are as follows. More than half of the respondents said that the resource of their information was insufficient (*n* = 309; 54%), a sufficient and good level of knowledge was indicated by 34% (*n* = 193) and 9% (*n* = 52) of the respondents, respectively. In contrast, the stock of their information at a very good level was assessed by only 1% of respondents (*n* = 8).

The overall level of knowledge of the students of the Silesian province regarding the importance of the intestinal microbiota in terms of colorectal cancer was subjected to statistical analysis taking into account sociodemographic data (age, place of residence), and the presence of chronic diseases. It was shown that students burdened with chronic diseases were characterized by significantly higher knowledge compared to healthy students (*p* = 0.001; *p* < 0.05). However, there was no significant effect of such determinants as age or place of residence of study participants (*p* = 0.08 NS).

Nearly half of the students surveyed at the University of Silesia in Katowice and the Silesian University of Technology (*n* = 160; 46.6%, *n* = 108; 46.4%, respectively) do not seek information on colon cancer as well as the gut microbiota. For respondents from the Medical University of Silesia, the figure settled at 27.1% (*n* = 29). Students were asked to give a subjective answer to the question-who should expand their knowledge of the gut microbiota and CRC? Respondents most often indicated a response referring to experts speaking in the mass media (*n* = 193; 33.8%), followed by general practitioners (*n* = 185; 32.4%) and teachers, as well as nutritionists (*n* = 98; 17.2%, *n* = 89; 15.6%, respectively). Nurses were the least frequently indicated group (<1%).

## 4. Discussion

Since 2010, the number of published papers on gut microbiota has been steadily increasing. Scientists are looking at links between the composition of the microorganisms residing in the gut and their impact on the health of the host. Gut dysbiosis can contribute to the development of obesity, carbohydrate disorders (such as diabetes), as well as cancer, mainly colon cancer. Probiotic supplementation in oncology patients appears to be a promising regimen to increase treatment efficacy, as well as reduce or inhibit tumor progression in colorectal cancer. Moreover, shortly, the study of gut microbiota may be part of screening efforts for lower gastrointestinal cancer. Knowledge and awareness of young adults about the gut microbiota, as well as the basics of oncology, including symptomatology along with diagnosis, can be an important aspect of the prevention of non-communicable diseases and cancer.

At the moment, the number of scientific publications on the topic of gut microbiota is small. However, several scientific articles have presented the results of surveys of respondents’ knowledge of probiotics, prebiotics, as well as colorectal cancer. A 2019 paper by Altamimi et al. [31] showed that more than half (*n* = 152; 56%) of the participating medical students from northern Jordan, with a mean age of 22.9 years (SD ± 1.33), knew the definition of probiotics. Significantly better results were obtained by Rahmah et al. [32] in a 2021 study, in which more than 90% of students in the health sciences department (*n* = 79) at Padjadjaran University (Indonesia) correctly handled a question relating to knowledge of the term probiotic. A 2019 study by Fijan et al. [33] on a population of health care professionals found that as many as 82.2% of respondents knew the correct definition of probiotics. On the other hand, the analysis of the answers given by the respondents surveyed in the present study showed that the students of the Silesian Medical University were characterized by a high level of knowledge of the topic under discussion (*n* = 93; 87%), and their average age was 21.8 years (SD ± 2.13).

A study by Sharma et al. [34] among students at Delhi University (India) found that 88.7% (*n* = 176) of respondents had information on probiotics. Less than 11% of them responded that preparations containing selected bacterial cultures benefit the host by promoting immunity. Similar results were obtained in a study by Jama-Kmiecik et al. [35], in which students at the Medical University of Wroclaw most often indicated that the benefit of consuming probiotics was improved immune system function. In our study, students from the Silesian region almost unanimously agreed that a proper composition of the intestinal microbiota has a beneficial effect on immune function (*n* = 526; 92%). In contrast, 72% of respondents (*n* = 402) were familiar with the definition of a probiotic.

In a 2019 study [34], Sharm et al. analyzed Indian students’ knowledge of prebiotics. Only a third of the students surveyed knew the correct definition (*n* = 66; 32.5%) of these products. The vast majority of respondents (*n* = 129; 63.5%) did not know the answer to the question. However, a higher percentage of respondents 54% (*n* = 89) familiar with the term prebiotic was demonstrated in the population of students from the Medical University of Wroclaw [35]. An even higher level of familiarity with the above term was reported in the present study (*n* = 370; 65%). It is worth mentioning that among 370 people declaring knowledge of the term prebiotic, as many as 90 were students from the Silesian Medical University in Katowice. Thus, as many as 84% of the surveyed students representing SUM demonstrated correct knowledge in this regard. In contrast, nearly 1/3 of the total number of surveyed students from universities in the Silesian province (*n* = 175; 31%) did not know the answer to the question on the definition of prebiotic and chose the option “don’t know.”

The intestinal microbiota performs many important functions in the human body. The correct composition of microorganisms inhabiting the terminal gastrointestinal tract prevents the settlement of pathogens, thus protecting against the development of many diseases, including metabolic diseases. A 2021 study by Barqawi et al. [36] found that more than ¾ of respondents (*n* = 321; 76.6%) were familiar with the term “microbiota.” The results of a study conducted on the population of residents of the United Arab Emirates are interesting against this background. This is because it was proven that respondents declaring membership in the medical industry had the highest level of knowledge in this area, giving >66% correct answers. In contrast, “non-medical” respondents gave only 18.7% correct answers. The demonstrated difference between the knowledge status of medical and non-medical survey participants was statistically significant (*p* < 0.001). A study, from the same year conducted in Jordan by Abu-Humaidan et al. [37], found that 39% of students (*n* = 157) demonstrated an advanced level of knowledge regarding microbiota. In contrast, basic knowledge of the topic covered was characterized by 50% (*n* = 202) of the respondents. Similar results were obtained in our study, in which a high level of knowledge was demonstrated among 62% (*n* = 66) of medical college students and in less than 17% (*n* = 77; 16.5%) of non-medical college students. The definition of “intestinal microbiota” was known by only 39% of respondents (*n* = 220). Only, one in four students demonstrated a wealth of information they possessed in the field covered (*n* = 143; 25%), while the majority of respondents had an average level of knowledge (*n* = 306; 54%).

The public needs to be aware of the symptoms that can herald the early stages of colorectal cancer. With the ability to react quickly, it is possible to detect the disease at an early stage, resulting in the rapid implementation of treatment and improved prognosis. Mhaidat et al. 2016 [38] surveyed university students located in Jordan. The students surveyed most often indicated that symptoms that could indicate the development of lower gastrointestinal malignancy primarily included abdominal pain (*n* = 567; 70.8%), the presence of a tumor (*n* = 557; 69.5%), weight loss (*n* = 449; 56.1%), rectal bleeding (*n* = 436; 54.4%) and blood in the stool (*n* = 224; 52.9%). An analysis of the data conducted by the authors of the present study showed a statistically significant difference between the number of correctly given answers in the area of knowledge of CRC symptoms and the type of university attended by the students surveyed (medical, non-medical). In a study by Ustundag et al. [39], health sciences students included blood in the stool (*n* = 750; 73%), rectal bleeding (*n* = 743; 72.3%), unintentional weight loss (*n* = 686; 66.7%) and fatigue (*n* = 656; 63.8%) among the most common symptoms of CRC. The problem of altered bowel frequency, including diarrhea and constipation, was indicated by 53.2% of respondents (*n* = 547). The self-reported study showed statistically significant differences between knowledge of CRC symptoms and university affiliation (*p* < 0.05). The best knowledge was characterized by students of the Medical University of Silesia in Katowice relative to those studying at the other two universities in Silesia. Participants in their own study also identified the presence of blood in the stool (*n* = 485; 85%), lower gastrointestinal bleeding (*n* = 453; 79%), diarrhea and constipation (*n* = 363; 64%, *n* = 356; 62%), as well as weight loss (*n* = 357; 63%) as the most common symptoms of CRC. Similar results were also obtained in a study by Pietrzyk et al. [40], which surveyed a population of students at the Medical University of Lodz. According to the respondents, symptoms heralding the development of colorectal cancer most often include such complaints as blood in the stool (80.6%), rectal bleeding (79.9%), and unintentional weight loss (73.9%). The authors of the study emphasize that the ratio of correct answers to the number of questions asked was higher among female respondents than among males. However, there is no doubt that students of medical schools, regardless of gender, should be equipped with the necessary knowledge of oncology at a high enough level to implement effective prevention, diagnosis, and therapy of cancer.

Pre-cancerous conditions include changes in the structure of the colon, which, if left untreated, most often lead to cancer. Screening plays an important role in detecting abnormalities in the lower gastrointestinal tract. Undoubtedly, pre-cancerous lesions can appear in anyone regardless of age and gender. Knowledge of the symptoms of colorectal cancer allows one to initiate diagnostic procedures in the early stages of cancer development. The chance of cure is also highest during this period. Medical school students (73.7%) who participated in the 2021 survey conducted by Aga et al. agreed with the above statement [41]. On the other hand, Chrobak-Bien et al. [42] proved that among respondents facing gastrointestinal disease as much as 88.5% (*n* = 177) had previously encountered the term “precancerous condition.” In contrast, less than 18% of them (*n* = 31) correctly identified diseases commonly considered to be factors that increase the risk of developing colorectal cancer. Nearly ¾ of the respondents (*n* = 147; 73.5%) were aware that CRC is more common in men than in women. Analysis of the responses given in the present study, by the surveyed students from the Silesian province, showed that 62% of the survey participants (*n* = 355) knew the definition of a precancerous condition, noting that the term refers to lesions that, if untreated, may contribute to the development of cancer in the future. Students surveyed for the present study were far less likely to perceive a cause-and-effect relationship between colorectal cancer incidence and gender. Nearly half of the respondents did not know the answer to this question (*n* = 238; 42%), while less than one in three respondents (*n* = 172; 30%) indicated the male gender. The difference between the results obtained in our study and those obtained by Chrobak-Bien [42] may be because the author’s study additionally provided an opportunity for students to declare their lack of knowledge in this area. In the study by Lewandowski et al. [43], it was found that almost half of the students surveyed (*n* = 79; 49.3%) declared that they had no clear opinion regarding the existence of a causal relationship between the presence of adenomatous polyps in the colon and the risk of developing cancer. In addition, 44% (*n* = 66) of the respondents believed that the type of diet used could affect the incidence of CRC. The in-house survey clearly emphasizes that university students from the Silesian province were characterized by a very good level of knowledge regarding the influence of dietary behavior on the magnitude of disease risk. More than 90% of respondents (*n* = 519; 91%) answered that diet is the most important factor determining the correctness of the colonic microbiota. In turn, the correct composition of the intestinal microbiota reduces the risk of lower gastrointestinal cancer. Nearly 50% of the students agreed with the above statement (*n* = 274; 48%).

Hrenchuk and Sidorchuk [44] 2022 conducted a study on young adults’ awareness of risk factors and preventive measures for colorectal cancer. Students in the study group were a distinct minority (*n* = 104; 19.3%). More than half of the respondents who claimed to have attended university showed a low level of knowledge (*n* = 69; 66.3%). Only less than 6% of respondents (*n* = 8) were characterized by high awareness of the issues covered. Imran et al. [45] 2016, on the other hand, compared the level of knowledge regarding colorectal cancer, among medical and non-medical students. Respondents who declared their affiliation with universities related to medical and health sciences showed a better level of knowledge than students from universities with other educational profiles (*p* < 0.001). Similar results were obtained in the author’s study, in which a statistically significant correlation was noted between the student’s level of knowledge regarding microbiota and colorectal cancer and the university from which they came. Students of the Medical University of Silesia were characterized by better knowledge compared to students of the Silesian University of Technology and the University of Silesia (*p* < 0.001). It is worth mentioning at this point that, taking into account the total surveyed population of students from Silesian universities, only 2% of them (*n* = 9) had a high level of knowledge in the field of colorectal cancer. In addition, the average value of the ratio of the number of correct answers given relative to the number of questions asked was only 50%, which indicates the low level of knowledge of the general studied population of students.

## 5. Strengths and Limitations

At the time of editing this paper, online database search engines displayed access to fewer than a handful of papers relating to the level of microbiota knowledge of various segments of the population. However, there is no doubt that a particular object of interest to researchers around the world is awareness, especially among young adults, of cancer, including colorectal cancer. Thus, shortly the number of published research papers on this topic may increase significantly.

## 6. Conclusions

The conducted study allowed us to formulate the following conclusions:As indicated by the results of the study, the awareness of the population of young adults about the microbiota in terms of colorectal cancer in the example of students from the Silesian province is unsatisfactory.The highest level of awareness of microbiota and CRC was demonstrated by students of the Silesian Medical University in Katowice. A significantly lower level of knowledge was characterized by the surveyed students of the University of Silesia and the Silesian University of Technology. The differences shown were due to the presence of issues related to microbiota and colorectal cancer in the context of medical education.There was also a significantly higher awareness of the microbiota in the aspect of colorectal cancer among respondents burdened with chronic diseases relative to healthy study participants.There was no significant effect of variables such as place of residence, age, or BMI value on the awareness of the young adult population about the microbiota in terms of colorectal cancer.It is worth considering the possibility of more educational activities aimed at young adults in the prevention of cancer, including colorectal cancer, especially in the population without medical education.

The results obtained in the present study identified the problem of low, unsatisfactory levels of knowledge in the field of microbiota and colorectal cancer among non-medical students from the Silesian province. Therefore, it is extremely important to educate the younger generation, regardless of their chosen field of study, in such a way that the knowledge they acquire allows them to make healthy choices daily. It is important to form correct habits in society in terms of both proper nutrition, but also physical activity, or avoiding stimulants such as alcohol and tobacco.

## Figures and Tables

**Figure 1 nutrients-14-04185-f001:**
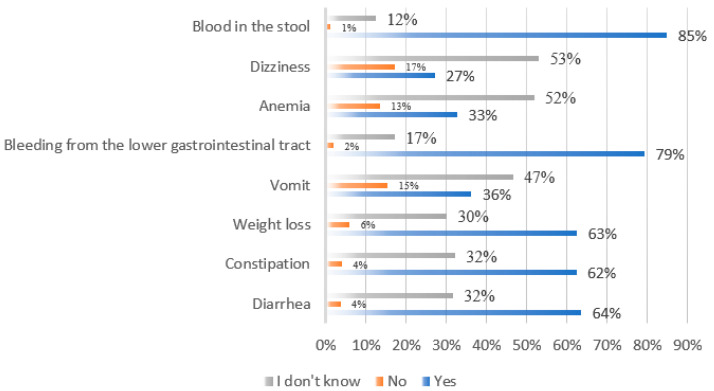
Knowledge of symptoms of colorectal cancer in the studied population of students from the Silesian province. *Source: own study.*

**Figure 2 nutrients-14-04185-f002:**
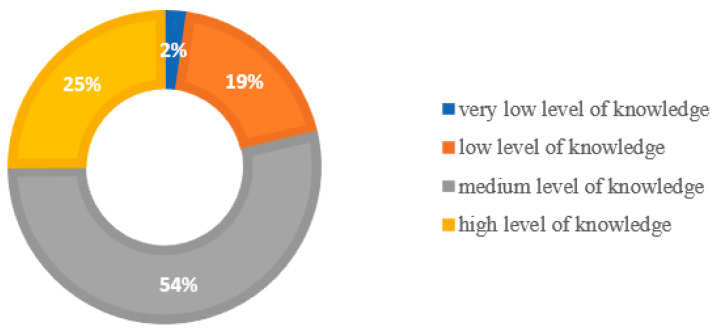
The level of knowledge of students from the Silesian province in the field of intestinal microbiota and colorectal cancer. *Source: own study.*

**Table 1 nutrients-14-04185-t001:** Characteristics of the studied group of students from the Silesian province. *Source: own study*.

Variables (*n*—Number of Subjects (%))		University ^1^	*n*—Number of Results (% of *n*^2^)
**Gender**	Women 437 (77%)	US	234 (83%)
		PolSL	116 (64%)
		SUM	87 (81%)
	Men 134 (23%)	US	49 (17%)
		PolSL	65 (36%)
		SUM	20 (19%)
**Residence**	City 441 (77%)	US	232 (82%)
		PolSL	132 (73%)
		SUM	77 (72%)
	Village 130 (23%)	US	50 (18%)
		PolSL	50 (27%)
		SUM	30 (28%)
**Body Mass Index (BMI)**	Underweight 65 (11%)	US	39 (14%)
		PolSL	10 (6%)
		SUM	16 (15%)
	Normal weight 364 (64%)	US	175 (62%)
		PolSL	118 (65%)
		SUM	71 (66%)
	Pre-obesity 105 (18%)	US	56 (20%)
		PolSL	33 (18%)
		SUM	16 (15%)
	Obesity I° 21 (4%)	US	7 (3%)
		PolSL	11 (6%)
		SUM	3 (3%)
	Obesity II° 6 (1%)	US	0 (0%)
		PolSL	6 (1%)
		SUM	0 (0%) 0 (0%)
	Obesity III° 3 (1%)	US	2 (1%)
		PolSL	1 (1%)
		SUM	0 (0%)
**Chronic diseases**	Yes 114 (20%)	US	58 (20%)
		PolSL	37 (20%)
		SUM	19 (18%)
	No 457 (80%)	US	225 (80%)
		PolSL	144 (80%)
		SUM	88 (82%)

^1^ US—the University of Silesia in Katowice, PolSL—Silesian University of Technology, SUM—Medical University of Silesia, ^2^ For N_US_ = 283; N_PolSl_ = 181; N_SUM_ = 107.

**Table 2 nutrients-14-04185-t002:** The number of correct answers in the field of microbiota depends on the university where the respondents studied. (Single-choice questions). *Source: own study.*

Question	University ^1,2^	The Number of Correct Answers *n* (%)	All of the Correct Answers (%)
What is microbiota?	US	88 (31%)	220 (39%)
	PolSL	60 (33%)	
	SUM	72 (67%)	
*p-value—Kruskal Wallis test **p*** ** = 0.0001**			
What is a probiotic?	US	195 (69%)	409 (72%)
	PolSL	121 (67%)	
	SUM	93 (87%)	
*p-value—Kruskal Wallis test **p*** ** = 0.005**			
What is a prebiotic?	US	172 (61%)	370 (65%)
	PolSL	108 (60%)	
	SUM	90 (84%)	
*p-value—Kruskal Wallis test **p*** ** = 0.0001**			
What factors have a beneficial effect on the gut microbiota?	US	190 (67%)	420 (74%)
	PolSL	128 (71%)	
	SUM	102 (95%)	
*p-value—Kruskal Wallis test **p*** ** = 0.0001**			
Please indicate the most important factor affecting the regularity of the gut microbiota.	US	261 (92%)	517 (91%)
	PolSL	161 (89%)	
	SUM	95 (89%)	
*p-value—Kruskal Wallis test p =* 0.37 *NS (not statistically significant)*			

^1^ US—the University of Silesia in Katowice, PolSL—Silesian University of Technology, SUM—Medical University of Silesia, ^2^ For N_US_ = 283; N_PolSl_ = 181; N_SUM_ = 107.

**Table 3 nutrients-14-04185-t003:** Number of correct answers given by respondents to questions about colorectal cancer depending on the university from which they came. (Single-choice questions). *Source: own study.*

Question	University ^1^	The Number of Correct Answers (*n*)	% of *n* ^2^	All of the Correct Answers (%)
What is a precancerous condition of the colon?	US	165	58%	355 (62%)
	PolSL	101	56%	
	SUM	89	83%	
*p-value—Kruskal Wallis test **p*** ** = 0.0001**				
Is the incidence of colorectal cancer gender-specific?	US	70	25%	172 (30%)
	PolSL	35	19%	
	SUM	67	63%	
*p-value—Kruskal Wallis test **p*** ** = 0.0001**				

^1^ US—the University of Silesia in Katowice, PolSL—Silesian University of Technology, SUM—Medical University of Silesia, ^2^ For N_US_ = 283; N_PolSl_ = 181; N_SUM_ = 107.

**Table 4 nutrients-14-04185-t004:** Students’ level of knowledge of gut microbiota and CRC depends on the university they attend. *Source: own study.*

University	Level of Knowledge	*n*—Number of Results	% of *n*
**University of Silesia in Katowice**	Very low	5	2%
	Low	66	23%
	Medium	166	59%
	High	46	16%
**Silesian University of Technology**	Very low	8	4%
	Low	39	22%
	Medium	103	57%
	High	31	17%
**Medical University of Silesia**	Very low	0	0
	Low	4	4%
	Medium	37	34%
	High	66	62%
*χ^2^ NW* = 96.51079	***p-value* = 0.0001**		*Vc (Cramér's V)* = 0.3

## Data Availability

Not applicable.

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
