# Peer review of "Colorectal Cancer and the Role of the Gut Microbiota—Do Medical Students Know More Than Other Young People?—Cross-Sectional Study"

_nutrients, 2022, doi:10.3390/nu14194185_

Round 1

Reviewer 1 Report

The paper compares knowledge of gut microbiota and link to CRC among students of 3 universities, one medical, 2 non-medical. Not too much of a surprise, the non-medical students did poorly compared to the medical students.  Whether this really represents a failure of education is somewhat questionable.  It really relates to the goals of the students and the course structure of the non-medical schools relative to those goals.   How and when information on microbiota and CRC would be taught in a liberal arts or technically oriented school is a bit difficult to figure out.    

Perhaps an additional point of the survey should be to query non-medical students as to the basis of their information, whether it had anything to do with university education or occurred outside?  And, how to integrate these sources into university coursework or social activities.

Author Response

Dear Reviewer,

We sincerely thank you for taking the trouble to review our manuscript. We are grateful for every comment that helped us improve our study.

„Perhaps an additional point of the survey should be to query non-medical students as to the basis of their information, whether it had anything to do with university education or occurred outside?  And, how to integrate these sources into university coursework or social activities.”

Authors add to the text more information’s (320-328).

We hope that the changes you have made will be satisfactory and sufficient for acceptance of our manuscript for publication in Nutrients.

With best regards, Authors

Reviewer 2 Report

Introduction

1.     Use aim of study but not hypotheses 

2.     Only one aim: Medical students knowledge of association between gut microbiota and CRC

3.     In my opinion, the introduction should be reduced. 

Materials and Methods 

Design should be added (crossectional) 

Use statistical analysis but not compilation 

Introduce again medical students. Had all student similar level of formation? Course. 

Results 

Table 1 Baseline characteristics. Table 2 and 3 : showed as n/%

Author Response

Dear Reviewer,

We sincerely thank you for taking the trouble to review our manuscript. We are grateful for every comment that helped us improve our study.

The authors decelerate that they have responded to the above comments. They have removed hypotheses, focused on a one aim, and shortened the introduction.

We added that it was a cross-sectional study. Statistical compilation replenished.

We hope that the changes you have made will be satisfactory and sufficient for acceptance of our manuscript for publication in Nutrients.

With best regards, Authors

Round 2

Reviewer 1 Report

The reply to my comments is adequate.